# Therapeutic Effects of Crocin Alone or in Combination with Sorafenib against Hepatocellular Carcinoma: In Vivo & In Vitro Insights

**DOI:** 10.3390/antiox11091645

**Published:** 2022-08-25

**Authors:** Suzan Abdu, Nouf Juaid, Amr Amin, Mohamed Moulay, Nabil Miled

**Affiliations:** 1Department of Biological Sciences, University of Jeddah, Jeddah 23445, Saudi Arabia; 2Biology Department, UAE University, Al Ain 15551, United Arab Emirates; 3The College, The University of Chicago, Chicago, IL 60637, USA; 4Embryonic Stem Cell Research Unit, King Fahd Medical Research Center, King Abdul Aziz University, Jeddah 22252, Saudi Arabia; 5Functional Genomics and Plant Physiology Research Unit, Higher Institute of Biotechnology Sfax, University of Sfax, BP261 Road Soukra Km4, Sfax 3038, Tunisia

**Keywords:** crocin, sorafenib, HCC, treatment, inflammation

## Abstract

This study investigated the therapeutic effects of the phytochemical crocin alone or in combination with sorafenib both in rats chemically induced with hepatocellular carcinoma (HCC) and in human liver cancer cell line (HepG2). Male rats were randomly divided into five groups, namely, control group, HCC induced group, and groups treated with sorafenib, crocin or both crocin and sorafenib. HCC was induced in rats with a single intraperitoneal injection of diethylnitrosamine (DEN), then 2-acetylaminofluorene (2-AAF). The HCC-induced rats showed a significant decrease in body weight compared to animals treated with either or both examined drugs. Serum inflammatory markers (C-reactive protein (CRP); interleukin-6 (IL-6); lactate dehydrogenase (LDH), and oxidative stress markers were significantly increased in the HCC group and were restored upon treatment with either or both of therapeutic molecules. Morphologically, the HCC-induced rats manifested most histopathological features of liver cancer. Treatment with either or both of crocin and sorafenib successfully restored normal liver architecture. The expression of key genes involved in carcinogenesis (TNFα, p53, VEGF and NF-κB) was highly augmented upon HCC induction and was attenuated post-treatment with either or both examined drugs. Treatment with both crocin and sorafenib improved the histopathological and inflammation parameters as compared to single treatments. The in vivo anti-cancer effects of crocin and/or sorafenib were supported by their respective cytotoxicity on HepG2 cells. Crocin and sorafenib displayed an anti-tumor synergetic effect on HepG2 cells. The present findings demonstrated that a treatment regimen with crocin and sorafenib reduced liver toxicity, impeded HCC development, and improved the liver functions.

## 1. Introduction

Liver cancer is the sixth most commonly diagnosed cancer and the fourth leading cause of cancer-related death worldwide, after lung, colorectal, and stomach cancers [1]. Liver cancer is a highly fatal disease, with most cases detected at late stages. Hepatocellular carcinoma (HCC) represents about 75–85% of primary liver cancers [2]. HCC is the world’s second leading cause of cancer deaths [3], characterized by a male predominance in incidence. HCC is characterized by considerable phenotypic and molecular heterogeneity. Recurrence following surgical treatment is the main cause of death for HCC [4].

Hepatitis viruses and chronic liver diseases, in particular cirrhosis, are the main risk factors for HCC. The etiology of liver diseases is rapidly changing due to improvements in the prevention and treatment of infections by Hepatitis B virus (HBV) and Hepatitis C virus (HCV) and to the rising incidence of metabolic syndromes such as non-alcoholic fatty liver disease (NAFLD) related to obesity. NAFLD is now recognized as a rapidly increasing cause of cirrhosis and HCC [5].

Over the past decade, improvements have been made in nonpharmacologic and pharmacologic therapies for HCC treatment. Nonpharmacologic therapies include liver resection, liver transplantation, trans arterial chemoembolization (TACE), and ablation. On the other hand, small-molecule targeted drugs such as sorafenib and lenvatinib, and monoclonal antibodies such as nivolumab are mainly used for systematic treatment of advanced HCC. In addition, several potential preclinical surgical or adjuvant therapies are being considered, including oncolytic viruses, mesenchymal stem cells, circadian clocks, gut microbiome composition, and peptide vaccines. All of these demonstrate varying degrees of inhibition of HCC [6,7]. As several potential anti-HCC drugs have been reported, many promising therapeutic targets are linked to taxonomic signaling pathways, including epigenetics, cell cycle, tyrosine kinases, etc., that influence HCC progression are also found [8].

Hypervascularity, vascular defects and angiogenesis, are often associated with HCC [9] and therefore offer critical targets for systemic treatments [10]. The Food and Drug Administration (FDA) approved therapy for patients with advanced-stage HCC using sorafenib [11]. Sorafenib approval in 2007 represented a breakthrough in treating HCC. It has been used since then as a standard treatment. Sorafenib prolonged median survival and time to progression by nearly 3 months in patients with advanced HCC [11]. It was the first systemic agent to demonstrate a survival benefit in advanced HCC patients. Sorafenib is a kinase inhibitor reducing mainly cell proliferation and angiogenesis [12]. It was reported that treatment with sorafenib had no effects on the cell cycle distribution of many tumors cell lines (HT29, Bax−/− HCT116, H460, and SKOV3) [13]. Sorafenib however delayed the cell cycle G1 phase in PC3 and T98G malignant glioma cells [13,14]. The ability of sorafenib to arrest the cell cycle may be related to Raf kinase inhibition. In irradiated cancer cell lines, sorafenib delayed the cell cycle G1 phase through modulating the expression of cyclin D1 and Rb genes [13]. Genes involved in cell cycle regulation (CDC45L, CDC6 and CDCA5) were downregulated by sorafenib in human HepG2 and Huh7 HCC cell lines [15]. In thyroid carcinoma cells, sorafenib was reported to inhibit multiple intracellular signaling pathways such as RAF-MAP kinase, leading to an increased proportion of cells in subG1 peak, cell cycle arrest and the initiation of apoptosis [16]. Furthermore, sorafenib induced cell cycle arrest and apoptosis in MV4–11, EOL-1 and NB4 leukemia cell lines [17,18]. Nevertheless, other studies reported that treatment of HepG2 HCC cell lines with sorafenib decreased the number of cells in G1 and increased cells in S phase [19]. Seemingly, the effect of sorafenib on the cell cycle is depending on the cellular background. Recently, additional multi-kinase inhibitors have been approved for patients with advanced HCC, including lenvatinib as a first-line therapy [20] and regorafenib as a second-line agent [21]. These drugs confer a median survival of less than 1 year for advanced HCC [22]. However, HCC is known to display high resistance to chemotherapy [23].

Treatment of liver cancer should not be limited to destroying tumor cells. The remodeling of the tumor microenvironment using a targeted treatment can be a significant advancement in treating liver cancer. Angiogenesis is a very important cancer-related process. Several angiogenic pathways were established as dysregulated in HCC, indicating that they may contribute to the development and morbidity of HCC. HCC development and evolution involve complex cross-talking signaling pathways such as cell proliferation, apoptosis, and angiogenesis.

Due to the high recurrence of HCC, there is a crucial need to develop novel preventive or therapeutic approaches that specifically target key molecules whose expressions are dysregulated during hepatocarcinogenesis. This would enhance patients’ survival, reduce the side effects of sorafenib and minimize treatment costs. It is also necessary to diversify diagnosis and prognostic methods to detect the disease at early stages and increase the chance of recovery.

Discovering appropriate new therapeutic molecules displaying an improved anti-HCC impact and reduced drawbacks is an important challenge. Natural products represent the main sources for developing/discovering novel anti-cancer or adjuvant chemotherapeutics. The use of natural products has the advantage of reduced side effects due to their well-known low toxicity. Phenolic compounds and carotenoids were also reported to display anti-cancer and hepato-protective capacities [24]. Crocin is a catenoid derived from saffron; stigmas of the flower of *Crocus sativus* [25], whose beneficial effects against cancer have been well-documented both alone [26,27] and encapsulated in nanoparticles [28]. Recent findings uniformly showed that saffron and its derivatives can affect carcinogenesis in a variety of in vivo and in vitro models. Particularly, crocin displayed a significant anticancer activity in breast, lung, pancreatic and leukemic cells [29]. In addition to its chemo preventive effects against HCC [30], crocin displayed anti-proliferation effects on cancer HepG2 cell lines [31] and contributed to autophagic HCC apoptosis [32]. The anti-cancer effect of crocin was explained by many mechanisms including the interaction with telomeric quadruplex sequences and down regulation of the hTERT gene. Data indicated that telomerase activity of HepG2 cells decreases after treatment with crocin, which is likely due to a down-regulation of the expression of the enzyme catalytic subunit [33]. Crocin was also found to induce apoptosis through chromatin condensation and DNA fragmentation in human pancreatic cancer cell line BxPC-3 [34].

In the present study, the therapeutic effects of crocin alone and in combination with sorafenib were investigated both in vitro and in a chemically induced-HCC animal model.

## 2. Materials and Methods

### 2.1. Animals, Hepatocarcinogenesis Induction and Treatment

The experimental design using the HCC induced rat model is presented in Figure 1. We developed a chemically induced-HCC animal model (Animal research ethics approval number # 677-20). In this model, rats (35 male albino Wistar rats weighing 160–200 g) were obtained from the King Fahd medical research center and the induction protocol described by [30] was followed with some modifications. HCC was initiated by a single intraperitoneal injection of DEN (Diethylnitrosamine) at a dose of 200 mg/kg body weight as previously described [30] dissolved in a normal saline solution. After initiation, all rats were subjected to 3 days of fasting followed by one day of re-feeding, as a mitotic proliferative stimulation. One week after DEN treatment, rats received 6 intra-gastric doses of 2-AAF (2-acetylaminofluorene) (30 mg/kg in 1% Tween 80) then one 2-AAF dose every week for 4 weeks, for promoting hepatocarcinogenesis. After 6 weeks of HCC induction, treatment lasted for another 6 weeks.
Group 1: Non-induced rats (control group).Group 2: HCC induced rats.Group 3: HCC induced rats treated with crocin.Group 4: HCC induced rats treated with sorafenibGroup 5: HCC induced rats treated with crocin/sorafenib combination.

For the control group (Group 1), animals were administered with distilled water (5 mL/kg body weight) and were subjected to a single dose saline injection throughout the experimental period.

Sorafenib and crocin at 98% purity were obtained from Sigma Chemical Co., Missouri, USA. Sorafenib, crocin and crocin/sorafenib combination were administrated by gavage at daily doses of 50 mg/kg/day in 1% Tween 80 for crocin as previously described [35] and 7.5 mg/kg/day in 1% Tween 80 for sorafenib, as previously reported [36]. The same doses were applied in combined treatment where sorafenib was administered first followed by crocin at 2 h interval time.

After one week of DEN treatment followed by six weeks of 2-AAF administration, rats were subjected to anti-cancer treatment during six weeks. At week 13, blood samples were collected through retro orbital punctures and animals were then sacrificed.

The body weights of rats were weekly recorded. Ratio of total body weight changes (%) = (Final weight–Original weight)/Original weight × 100.

Liver index (%) = Liver weight/Final body weight × 100.

### 2.2. Evaluation of Serum Biochemical Parameters

Blood samples were collected at the time of sacrifice in serum tubes (vacutainer) and were centrifuged at 3000 rpm for 20 min at 4 °C to obtain serum. The serum levels of aspartate aminotransferase (AST), alanine aminotransferase (ALT), and alkaline phosphatase (ALP), total protein (TP), total cholesterol (TC), triglycerides (TG), Conjugated bilirubin, C-reactive protein (CRP), interleukin-6 (IL-6); lactate dehydrogenase (LDH) and Protein induced by vitamin K absence-II (PIVKA-II) were all estimated using the rat ELISA kits obtained by BioSource USA following the instructions and steps contained in the internal kits bulletin.

### 2.3. Evaluation of Liver Homogenate Biomarkers

Liver tissues were homogenized using buffer (1:10, *w*/*v*) containing 100 mM KCl, 100 mM potassium buffer (pH 7.4), and 1 mM EDTA for 90 s and then homogenates were centrifuged at 10,000× *g* for 30 min at 4 °C to get supernatants. The supernatants were used to measure the concentration of the glutathione (GSH) and the liver cancer biomarker α fetoprotein (AFP) using ELISA rat kits purchased from (BioSources, San Diego, CA, USA) and following the kit instructions. Protein concentration of the above supernatant was estimated by method of Lowry et al. [37].

### 2.4. Histopathological Staining

Liver tissue samples fixed in 10% neutral-buffered formalin solution were dehydrated in ethanol, cleared in xylene, and impeded in paraffin to form tissue blocks. The latter were sectioned (4–5 μm), and the slides were either stained by hematoxylin and eosin (H & E) or used for immunostaining.

### 2.5. Immunohistochemistry

Immunohistochemical staining Ki67 had been performed using Benchmark GX machine with Standard Cell Conditioning (CC1) on the 4 µm paraffin sections mounted on positively charged slide incubated for 1 h at 60 °C. Sections were incubated for 16 min, 37 °C with the pre-diluted primary antibody, rabbit anti-Ki-67 (clone 30-9). The antigen-antibody reaction was detected with 3,3-diaminobenzidine (DAB) using an Ultraview Universal Dab Detection Kit. The sections were counterstained by Hematoxylin for 4 min, dehydrated in a graded series of ethanol, cleared with xylene and cover slipped with DPX.

### 2.6. Molecular Analysis by Real-Time PCR

Total RNA was isolated from tissue samples stored at −80 °C in RNA later using an RNA extraction Kit (Cat No.: R1200 Solar bio, Beijing, China). The cDNA was synthesized by reverse transcription of RNA using Quantiscript reverse transcriptase according to the manufacturer’s instructions (Cat No.: RP1100, Solar bio, China). Specific primers for tumor necrosis factor alpha (TNF-a) (5′CCCTGGTACTAACTCCCAGAAA-3′, 5′TGTATGAGAGGGACGGAACC-3′, [38]), vascular endothelial growth factor (VEGF) (5′ACAGAAGGGGAGCAGAAAGCCCAT-3′, 5′CGCTCTGACCAAGGCTCACAGT-3′, [39]), tumor suppressor (p53) (5′CCTATCCGGTCAGTTGTTGGA-3′, CCTATCCGGTCAGTTGTT-GGA-3′, [40]) and nuclear factor kappa-light-chain-enhancer of activated B cells (NFκB) (5′GCAAACCTGGGAATACTTCATGTGACTAAG-3′, 5′ATAGGCAAGGTCAGAATGCACCAGAAGTCC-3′, [41]) and GAPDH (5′CAACTCCCTCAAGATTGTCAGCAA-3′, 5′GGCATGGACTGTGGTCATGA-3′, [42]) genes were chosen. Quantitative real-time PCR (qPCR) was performed using QuantiTect SYBR Green qPCR Master Mix in a StepOnePlus Real-Time PCR system (Applied Biosystems, San Francisco, USA) and reaction cycles (Reaction mixtures were incubated for 10 min at 95 °C, followed by 40 cycles of 15 s at 95 °C, 1 min at 60 °C, and, finally, 15 s at 95 °C, 1 min at 60 °C, and 15 s at 95 °C). The quantities of the critical threshold (Ct) of target genes were normalized with quantities (Ct) of the internal control (GAPDH). Levels were expressed relative to normal control samples.

### 2.7. Cancer Cell Line

Hepatocellular carcinoma cell line (HepG2) was provided from American Type Culture Collection (ATCC) (Manassas, VA, USA). The cell line was cultured in DMEM medium supplemented with 10% fetal bovine serum (Sigma Aldrich, St. Louis, MO, USA), containing 1% of 100 U/mL penicillin and 100 μg/mL streptomycin (Sigma Aldrich), and incubated at 37 °C in a humidified 5% CO2 atmosphere. Cells were sub-cultured each 3 days using trypsin 0.25%.

### 2.8. Cytotoxicity Assay

The anticancer activity of Crocin and Sorafenib on HepG2 cells was determined by the MTT (3-(4,5-dimethyl thiazol-2yl)-2, 5-diphenyl tetrazolium bromide) assay used to assess the cytotoxicity [43]. Cells (10,000/well) were plated in 96-well plates in 180 μL of complete growth medium. The attached Cells were treated with different concentrations of Crocin (100 μM, 150 μM, 200 μM, 250 μM and 300 μM) and Sorafenib (5 μM, 10 μM, 20 μM, 30 μM and 40 μM) or sorafenib/crocin equimolar combination (final concentration of the mix is 6.25 μM, 12.5 μM, 25 μM, 50 μM or 100 μM) for 48 h. After incubation, the medium was removed carefully from each well and washed with 90 μL of fresh culture medium before adding 10 µL of 3-[4,5-dimethylthiazol-2-yl]-2,5-diphenyltratrazolium bromide (MTT)) (Cat No.: M1020, Solar Bio, China) solution and continuing the culture for 4 h in 5% CO2 incubator. After that, 1 mL of DMSO (solubilizing reagent) was added to each well and mixed then incubated for 45 s. The presence of viable cells was visualized by the development of purple color due to formation of formazan crystals. The suspension was transferred to the cuvette of a spectrophotometer and the OD (optical density) values were read at 570 nm by using DMSO as a blank. Measurements were performed and the concentration required for a 50% inhibition of viability (IC50) was determined graphically by plotting concentration of the drug in X axis and relative cell viability in Y axis.
Cell viability (%) = Mean OD/Control OD × 100%

For combined treatment using sorafenib and crocin, the combination index (CI) was calculated from the formula:
(1)$$CI=\frac{IC50\;\mathrm{of}\;\mathrm{sorafenib}\;\mathrm{combination}}{IC50\;\mathrm{of}\;\mathrm{sorafenib}\;\mathrm{alone}}+\frac{IC50\;\mathrm{ofcrocin}\;\mathrm{combination}}{IC50\;\mathrm{of}\;\mathrm{crocin}\;\mathrm{alone}}$$

The nature of drug interaction is defined as synergism if CI < 0.8; antagonism if CI > 1.2; and additive if CI is in the range 0.8–1.2 [44].

### 2.9. Cell Cycle Analysis

The effect of the crocin and sorafenib on the cell cycle distribution of HepG2 cells was determined by using the flow cytometry analysis (FCA) and the Cell Cycle Analysis Kit (Cat No.: CA1510, Solar bio, China). The cells were subjected to a treatment with the free media (control) or with the pre-determined IC50 of crocin, sorafenib or crocin/sorafenib combination for 48 h. After incubation, the cells were collected by trypsinization and washed twice with ice-cold PBS then re-suspended in 0.5 mL of PBS. Two milliliters of 60% ice-cold ethanol were added, and cells were incubated for fixation at 4 °C for 1 h. Upon analysis, the fixed cells were washed and resuspended in 1 mL of PBS containing 50 μg/mL RNase A and 10 μg/mL propidium iodide. After 20 min of incubation in darkness at 37 °C, the cells were analyzed for DNA content using FL2 (λex/em 535/617 nm) signal detector (ACEA Novocyte™ flow cytometer (ACEA Biosci-ences Inc., San Diego, CA, USA). About 12,000 events occurred per sample. Cell cycle distribution was calculated using ACEA Novo-Express™ software (ACEA Biosciences Inc., San Diego, CA, USA).

### 2.10. Apoptosis Assays

The Effect of (Crocin, Sorafenib, and Crocin-Sorafenib) on apoptosis and necrosis of the studied cell lines was determined by flow cytometry analysis using Annexin/V-FITC apoptosis detection kit, according to the manufacturer’s instructions (Annexin V-FITC/PI Apoptosis Detection Kit, CA1020, Solarbio, Beijing, China). Briefly, the cell line was treated with the respective IC50 of the crocin, sorafenib or both crocin and sorafenib for 48 h. Subsequently, the cells were collected by trypsinization, washed twice with ice-cold PBS, and re-suspended in 0.5 mL of annexin/V-FITC/PI solution for 30 min in dark according to the manufacturer’s protocol. After staining at room temperature, the cells were injected into the ACEA Novo-cyte™ FCA (ACEA Biosciences Inc., San Diego, CA, USA) and analyzed for FITC and propidium iodide fluorescent signals using FL1 and FL2 detectors, respectively (λex/em 488/530 nm for FITC and λex/em 535/617 nm for PI). About 12,000 events were acquired and positive FITC and/or PI cells was quantified by quadrant analysis and calculated using ACEA NovoExpress™ software (ACEA Biosciences Inc., San Diego, CA, USA). Each treatment was repeated three times and data represents means ± SEM of three replicates.

### 2.11. Statistical Analysis

The data obtained during the study were analyzed utilizing IBM SPSS Statistics for Windows, version 23 (IBM SPSS, IBM Corp., Armonk, NY, USA). The Shapiro–Wilk test was utilized to evaluate normal value distribution. Collected value was presented as mean +/− standard deviation (SD). Statistical comparisons were made by one-vay analysis of variance (ANOVA) and Newman-Keuls was used as a post hoc test, followed by least significant difference (LSD) analysis. *p* value < 0.05 was considered statistically significant. The IC50 for all cell lines was determined by ED50 plus V1.0 software. All data were expressed as mean ± standard error (SEM) of three replicates (*n* = 3). Statistical data were analyzed by Prism (V5, Co., San Diego, CA, USA) and the differences between groups were considered significant at * *p* < 0.05.

## 3. Results

### 3.1. Effect of Crocin and Sorafenib Treatment on HCC Induced Rats

In the two-stage HCC induction model protocol, initiation and promotion steps are important in developing HCC where promotor induces clonal expansion of initiated cells [45]. To improve HCC development in animal models, exposure to a tumor promotor, such as 2-acetylaminofluorene (2-AAF), often helps inducing the formation of altered hepatocytes foci (AHF) and hyperplastic nodules that would ultimately develop into HCC [45,46]. Fasting-refeeding and employing 2-AAF after 2 weeks of using DEN are reported as mitotic proliferative stimuli [47]. In the present model, initiation is followed by a mitotic proliferative stimulus (fasting and re-feeding) during treatment with promoting agent such as 2-Acetyl Aminofluorene (2-AAF) that induces selective proliferation of the initiated cell population over non-initiated cells in the target tissue [48,49]. Although feed deprivation for three days is considered as a severe stress and itself may lead to body weight loss, it reduces the death that results from other models, such as cutting a piece of the liver, and we have established this model in our previous experiments to induce liver cancer [50,51].

#### 3.1.1. Effect of Crocin and Sorafenib on Liver and Body Weights

Changes in total body weights (TBWs) for the non-induced (Group 1) and HCC induced (group 2) rats were recorded from week 1 to week 6 of the experimental period (Table 1). TBWs were significantly (*p* < 0.05) decreased in the HCC rats as compared to the control group (Table 1). After 6 weeks post-HCC induction, the ratio of total body weight increase was 40.51% in the control group and only 28.16% in the HCC group. HCC induced a significant weight loss compared to the control.

Animal treatment started at week 7. The initial TBWs in HCC, G3, G4 and G5 groups were significantly decreased versus control group (G1) (*p* < 0.05) (Table 2). Meanwhile, the final TBWs were significantly increased upon treatment with crocin (G3), sorafenib (G4) or (sorafenib/crocin) combination (G5). The TBW increase was comparable to the normal group for all treatments and significantly higher than that of the non-treated group (*p* < 0.05) (Table 2). Anti-tumor treatments succeeded to restore the animals body weight at week 13. The liver index showed no significant changes in the different studied groups (*p* < 0.05). Meanwhile, as compared to the control animals (Table 2).

#### 3.1.2. Effect of Crocin and Sorafenib on Biochemical Parameters

Liver function tests were analyzed for the different studied groups (Table 3). The serum levels of ALT, AST, ALP, and conjugated bilirubin were significantly increased upon HCC induction (G2). Treatment with crocin and to a lesser extent with sorafenib or (sorafenib/crocin) combination could reduce the enzymatic activity to levels close to those of normal animals. The treatment with crocin was more effective in restoring the serum level of liver enzymes. Treatment with sorafenib partially reduced these activities whereas adding crocin to sorafenib improved its ability to recover serum enzymes normal levels.

The fact that HCC elevated liver enzymes ALT, AST, and ALP is indicative of a liver damage upon HCC induction. Treatment with crocin alone or combined with sorafenib reduced efficiently the elevated liver enzyme’s levels compared to sorafenib alone. A similar effect was observed for total protein concentration that increased for HCC induced animals and was restored totally for crocin treatment and partially for the other treatments.

Total cholesterol and triglycerides, CRP, LDH and IL-6 serum levels in HCC animals significantly increased indicating clear liver damage and inflammation induction. Crocin alone or combined with sorafenib markedly reduced concentrations of those markers compared to control levels. Administration of sorafenib alone significantly deceased the levels but in a less effective manner as compared to crocin or (sorafenib/crocin) mix.

Expectedly, the levels of the Serum tumor marker PIVKA-II, liver tumor marker AFP and oxidation stress enzymes (GSH and MDA) significantly increased upon HCC induction. The various treatments attenuated the levels of PIVKA-II that remained however higher than the control. Crocin or (sorafenib/crocin) combination displayed a better performance than sorafenib alone in attenuating tumor makers and oxidation parameters. The combination of crocin with sorafenib succeeded to lower conjugated bilirubin, LDH and AFP as compared to sorafenib alone (*p* < 0.05) (Table 3).

#### 3.1.3. Histopathological and Immunostaining Changes Induced by Crocin and Sorafenib

H & E stain of the control group displayed a normal liver structure. However, HCC animals exhibited a marked liver histological damage indicated by wide distribution of tumor cells and nodules, hemorrhage, angiogenesis, or vascular invasion, hypercellularity, and lymphocytic infiltration in many places (Figure 2). The tumor nodules were of trabecular and solid patterns with irregular demarcation, basophilic and coagulative cytoplasm. Many tumor nodules were vascularized. Clear cells variant of tumor nodules, cytologic atypia, mitotic figures, Mallory-bodies within the tumor cells, and unpaired arteries were noticed. Reduced number of portal triads and bile ducts were also noticed. Bile production was frequently observed. The clear-cell variant of HCC, characterized by clear cytoplasm was observed in some parts of the tumor. The tumor cells often have an increased cell size but show regular nuclei without atypia.

However, when rats were treated with crocin, sorafenib, or (sorafenib/crocin), the hypercellularity and tumor cells and nodules, as well as hemorrhage and vascular invasion were markedly reduced. In the crocin group particularly, the cytoplasm was eosinophilic, and the portal areas reappeared. The tumor nodules reduced in number and size with increased apoptotic cells. However, the sorafenib group distinctly exhibited extensive fat vesicles in the liver tissue. The cytoplasm was basophilic in many areas with marked reduction in tumor nodules. In the animals treated with the (sorafenib/crocin) the liver tissue exhibited color differentiation, marked reduction in tumor nodules with increased apoptosis and necrosis and decreased number of fat vesicles. Sinusoids reappeared between hepatocytes strands.

#### 3.1.4. Immunohistochemical Staining of Ki67

Ki-67 protein in the nucleus is associated with cell proliferation. In HCC patients, high levels of Ki-67 are usually indicative of tumor aggressiveness such as an advanced tumor stage [52]. Ki-67 was therefore proposed as an independent prognostic factor for surgically resected HCC [53]. Our results show increased Ki67 expression in HCC animals compared to the other groups (Figure 3).

Crocin treatment alone or in combination with sorafenib reduced significantly the ki-67 expression to levels close to those of the control animals (*p* < 0/05) (Figure 3). Sorafenib was less effective in reducing the ki-67 to normal levels (*p* > 0.05). The crocin/sorafenib mixture outperformed the single drug treatment in reducing the Ki-67 expression to significantly lower values (*p* < 0.05) as compared to the induced rats (Figure 3).

#### 3.1.5. Molecular Changes in Gene Expression Induced by Crocin and Sorafenib Treatment

Expression levels of TNFα, VEGF, p53 and NFκB were high for the HCC induced rats (Table 4). HCC induction was accompanied by an overexpression of TNFα, VEGF, p53 and NFκB as compared to non-induced rats (*p*-value < 0.05) indicating a high necrosis, cellular damage, and angiogenesis. These processes are characteristic of HCC development. The expression fold increase was higher than 100. This is to be explained by low expression yields for normal rats followed by a strong induction of these genes due to HCC. After treatment with sorafenib, the fold change dropped sharply for all studied genes but was still higher than 100 for TNFα and p53 and reached 54.57 and 12.87 for VEGF and NFκB, respectively. Crocin treatment of HCC was effective in reducing sharply target genes TNFα, VEGF, p53 and NFκB expression to fold changes of 1.17, 5.02, 0.99 and 4.21, respectively. Upon crocin treatment, expression levels were similar to those of normal animals (*p*-value > 0.05). Combined treatment with sorafenib and crocin yielded intermediate fold changes between single treatments. Crocin was more effective in attenuating the rising of target genes expression due to HCC induction.

VEGF is produced in HCC cells in concentrations that are usually correlated with tumor size and disease stage [54]. VEGF mediates angiogenesis by increasing the proliferation and differentiation of endothelial cells mediated also by the fibroblast growth factor [55]. Its overexpression upon HCC induction correlates with cancer development and node formation. The treatment with crocin alone or in combination with sorafenib was effective in reducing the VEGF gene expression. Combined treatment downregulated VEGF that reached expression levels close to those of normal animals (*p*-value > 0.05).

p53 is a transcription factor involved in cell-cycle regulation and apoptosis that was recognized as a tumor suppressor gene and the most frequently mutated in human cancer with a mutation rate of about 50% in human cancer cases [56]. In HCC, p53 has been also reported to promote autophagy and act as an antioxidant to prevent DNA damage and genomic instability by inhibiting mTOR signaling [57]. P53 is a pro-apoptotic protein that plays a key anti-tumor role. Under conditions of cellular stress and damage, p53 tends to prevent further damage by inducing cell cycle arrest to permit DNA repair or through apoptosis [58]. Treatment with crocin alone or in combination with sorafenib reduced the expression levels of p53 upon HCC induction (Table 5). Livers of sorafenib-treated rats displayed high levels of p53 expression as compared to those treated with crocin alone or in combination with sorafenib.

### 3.2. Effect of Crocin and Sorafenib Treatment on HepG2 Cell Viability and Cell Cycle

#### 3.2.1. Cytotoxicity

The cytotoxicity of sorafenib and crocin against human hepatocarcinoma, was evaluated by incubating the HepG2 cells with different doses of crocin (50 to 300 μM) and sorafenib (5 to 40 μM) for 48 h. After 48 h of incubation, cell viability was determined by the MTT assay. The half maximal inhibitory concentration (IC50) of sorafenib as close to 10 μM (Figure 4A) whereas that of crocin was around 200 μM (Figure 4B).

The cytotoxicity of an equimolar mix of crocin and sorafenib was assayed on HepG2 cells (Table 5). The mix concentration varied from 6.5 μM to 100 μM. IC50 of the mix sorafenib/crocin at 48 h was estimated to 12.51 μM. The calculated combination index (CI) value 0.65 was lower than 0.8 indicating a synergism between the two drugs [44]. The combination of sorafenib/crocin at 1:1 molar ratio displayed a more potent inhibitory power of the cancer cell viability than the individual drugs sorafenib and crocin.

#### 3.2.2. Cell Cycle

To investigate the potential anti-proliferative effects of sorafenib and crocin against cancer cells, the HepG2 cell lines were incubated for 48 h with sorafenib or crocin or their mix, at IC50 values. This will determine whether the antiproliferative effect was due to the arrest of the cell cycle at a specific phase. Cell cycle arrest in the growth phase by anti-tumor molecules leads to cell death by apoptosis [59]. Damaged cells through cell cycle arrest into G1 or G2/M phases undergo apoptosis usually due to subsequent aberrant mitosis [60]. This occurs usually as a late apoptosis resulting from several distinct pathways such as DNA damage, resulting in cell arrest in the G1 or G2/M phase [61]. Subsequently, the cell cycle phases were analyzed using DNA content flow cytometry (Figure 5).

While sorafenib did not display any cycle arrest, crocin and sorafenib/crocin mix exerted an anti-proliferative effect and arrested the cell cycle of the cancer cell lines in G1 phase (Figure 5). The crocin significantly increased cell population at the sub-G and G1 phases with reciprocal decrease in the S phase (Figure 5). In response to the (sorafenib/crocin) combination, cell population in the G1 phase significantly increased (Figure 5) with a concomitant decrease in the cell line in S phase, indicating a strong anti-proliferative effect as compared to non-treated cells.

These results suggest that both crocin and (sorafenib/crocin) combination exerted an anti-proliferative action through an apoptosis-inducing property. Sorafenib did not display a significant change in cell line populations in G1 and G2/M phases which indicate it exerts an anti-proliferative action through other mechanisms rather than cell cycle arrest.

#### 3.2.3. Apoptosis

To identify the form of cell death induced by sorafenib, crocin or (sorafenib/crocin), the tested cancer cell line was incubated for 48 h with the various anti-HCC molecules at IC50. Cells were then analyzed using the Annexin/V-FITC and subsequently, apoptotic, and necrotic cells were differentiated by flow cytometry apoptosis detection assay. Sorafenib, crocin and combined treatment significantly increased the number of necrotic and apoptotic cells (Figure 6).

Sorafenib induced cell death by necrosis (2.2%) early apoptosis (6.1%) and late apoptosis (11.6%) (Figure 6). The fact that sorafenib increased necrosis and apoptosis but did not induce cell cycle arrest is indicative that it acts through differs mechanisms such as inactivation of the RAF/MEK/ERK pathway HCC and receptor tyrosine kinases involved in tumor progression and angiogenesis [62].

Crocin increased cell death by necrosis (3.6%) and induced only late apoptosis death (6.5%). Interestingly, combining crocin with sorafenib increased cell necrosis to 6.8% and late apoptosis to 15.8% whereas death by early apoptosis remained low (0.9%). Adding crocin to sorafenib seemed therefore to accelerate cell death by necrosis and late apoptosis. Sorafenib, crocin and combined treatment significantly increased the number of necrotic and apoptotic cells. The cell death by apoptosis and necrosis confirmed the synergetic effects.

## 4. Discussion

*Crocus sativus* (saffron) is a rich source of medicinally important crocin, a naturally occurring carotenoid. The potential of crocin in cancer prevention and therapy were extensively investigated in recent years. The present study was designed to explore the efficacy of crocin alone or in combination with the classical drug sorafenib against HCC.

In the present study, results showed that treatment with crocin significantly reduced serum markers of liver functions which agrees with previous reports [63,64]. Improved levels of liver enzymes by crocin treatment might be due to its potent antioxidant effect and ability to maintain the integrity of hepatic cell membrane thereby preventing transfusion of these cellular enzymes into the serum [65].

The serum levels of Interleukin 6 (IL-6) that acts as a pro-inflammatory cytokine and liver inflammation c-reactive protein (CRP) were increased in HCC. This due to inflammation associated with HCC induction. These inflammation markers decreased upon treatment with crocin. This is in accordance with previous reports that crocin attenuated inflammation in rats [66]. Sorafenib reduced inflammation but to a lesser extent. The combination (sorafenib/crocin) was also effective in reducing inflammation.

Lactate dehydrogenase (LDH) is an indirect marker of tumor hypoxia, angiogenesis and poor prognosis in HCC [67]. LDH plays an important role in making the body’s energy. The present study results indicated significant elevation of LDH level in HCC and significant recovery in crocin and (sorafenib/crocin) groups suggesting crocin role in HCC treatment.

Alpha fetoprotein (AFP) is the most used marker for detecting HCC [68]. Prothrombin induced by vitamin K absence-II (PIVKA-II), considered as a potential biomarker that complements AFP for the diagnosis of HCC. In the present study PIVKA-II and AFP increased in HCC, which is in accordance with previous studies [68] reporting that (PIVKA-II) increased in the serum of HCC patients because of an acquired defect in the posttranslational carboxylation of the prothrombin precursor in malignant cells. The present study showed the efficacy of crocin and (sorafenib/crocin) treatments in reducing PIVKA-II and AFP levels indicating the anti-tumor activity of both formulations.

Previous studies have shown crocin prevented early liver damages of liver cancer [69], induced apoptosis [31] and autophagy [32]. Crocin was reported to play an important role in the regulation of angiogenesis pathways in breast cancer [70]. Various mechanisms for crocin to suppress cancer cell proliferation and induce apoptosis such as inhibition of key enzymes in nucleic acids synthesis, and remodification of epigenetic properties were suggested by [71]. In the present study, crocin apoptosis induction and growth arrest in HCC was confirmed. Likewise, it also previously suggested that crocin induces antitumor effect and cancer cell death in colorectal cancer cells through p53-dependent and independent mechanisms [27]. Crocin was also shown to exert anti-proliferative and apoptotic effects on cutaneous squamous cell carcinoma in vivo [72]. In the present study, crocin treatment alone or associated with sorafenib downregulated TNF-α, NF-κB, VEGF and p53 genes. Crocin is likely to exert an anti-tumor activity through downregulation of TNF-α and NF-κB. Likewise, Amin et al. [30] demonstrated that crocin treatment succeeded to reduce the expression of NF-κB and the activity of the TNF-α in HCC induced rats. TNF-α activates the extracellular signal-related kinase (ERK), c-Jun NH2-terminal kinase (JNK) and NF-κB, which are potent inducers of inflammation [73]. In addition to its pro-inflammatory role, TNF-α is an anti-tumor cytokine and was reported to activate NF-κB formation [74]. Nevertheless, many studies demonstrated the relationship between TNF-α and NF-κB and tumor development. Transfection of the human TNF-α gene into mice results in an increase in the number of hepatic metastases [75]. Furthermore, TNF-α was shown to induce the epithelial-to-mesenchymal transition in Human HCC cell lines [76]. It is noteworthy that high TNF-α expression levels were correlated with poor outcomes in HCC patients who received post-surgery adjuvant sorafenib [77]. This was explained by resistance to the sorafenib treatment. These results are similar to those obtained in the present work using an HCC induced rat model and showing high TNF-α expression levels when using sorafenib as compared to the treatment by crocin alone. The inhibition of TNF-α expression was therefore successfully used as an associated treatment to reduce sorafenib resistance and suppress the progression of HCC in vitro and in vivo [76]. Furthermore, NF-κB was identified as a potential regulatory hub that is inactivated by in the HCC human cell model [30]. The fact that crocin downregulated the VEGF gene recalls previous reports that crocin, metformin, and the combination treatment resulted in a dramatic decrease in the VEGF and MMP9 protein content in a mice induced breast cancer [78]. Increasing the p53 expression yield upon HCC induction is due to the stress stimulation of the “guard protein” expression direct to repair damages. This result is in line with previous reports that the expression levels of p53 increased in liver tissues for the HCC-induced rats as compared to the control ones [79]. Treatment with crocin or sorafenib/crocin and to a lesser extent with sorafenib reduced the p53 expression rates. This is to be explained by a reduction in the cellular damage, stress, and inflammation upon treatment with crocin alone or in combination with sorafenib.

Although clinical trials have confirmed the efficacy of sorafenib, the drug has several side effects and patients are rapidly develop resistance. Sorafenib increased the median survival time by approximately 3 months [80]. The main drawbacks of sorafenib are toxicity and increase oxidation and inflammation. The addition of crocin succeeded to attenuate oxidation, inflammation, and liver damage due to HCC induction and likely to the drug itself. Although the treatment with crocin reduced the liver tumor nodules, sorafenib alone or in combination with crocin yielded a marked reduction in tumor nodules with increased apoptosis and necrosis. Furthermore, the combination of crocin and sorafenib displayed fewer fat vesicles, as compared to sorafenib alone. Therefore, combing crocin with sorafenib in the treatment of HCC succeeded to reduce inflammation and oxidation and improve anti-tumor performance as compared to sorafenib alone. Although highly effective in reducing inflammation, oxidation, and liver damage, crocin alone displayed a lower histological amelioration of tumor aspects as compared to the combination (sorafenib/crocin). Furthermore, crocin in combination with sorafenib significantly inhibited the proliferation of HCC in liver tissue. Combined treatment with both crocin and sorafenib improved liver recovery and reduced the levels of tumor markers better than sorafenib or crocin alone. This is indicative of a synergetic action between both molecules. This fact was confirmed by in vitro analysis showing that the combined treatment inhibited proliferation and induced apoptosis in HepG2 cancer cell lines mainly through cell cycle arrest in G_0_ phase. A synergism was observed between the two components in vivo and in vitro. Combining sorafenib and crocin at an equimolar ratio was effective in reducing cell viability and increasing apoptosis as compared to the use of either dug alone, as it was reported for other sorafenib combined treatments [80]. The combined effect is likely to be explained by a synergism between sorafenib known to inhibit tumor proliferation and increase apoptosis in HCC [62]. and crocin. While sorafenib targets mainly the RAF/MEK/ERK pathway and receptor tyrosine kinases involved in tumor progression and angiogenesis, a complementary synergetic action of crocin may occur through other mechanisms including the interaction with telomeric quadruplex sequences and down regulation of hTERT expression, resulting in a decreased telomerase activity of HepG2 cells [33].

## 5. Conclusions

Crocin efficiently improved the induced inflammation, oxidation, and liver damage parameters. This is in line with its protective liver properties. Nevertheless, treatment with sorafenib alone or in combination with crocin evidently improved the microscopic liver pathology by reducing the degenerative changes and inhibiting the proliferation of cancer cells. A combined treatment was nevertheless more effective than that with sorafenib alone. The combination of crocin with sorafenib improved physiological parameters such as oxidation, inflammation and liver damage as compared to sorafenib alone. Through the improvement of inflammation and oxidation parameters and an anti-proliferative action, crocin improves the action of sorafenib. The combination of sorafenib and crocin is a promising treatment for HCC that reduces cell damage and improves the efficiency of sorafenib in reducing cell proliferation and carcinogenesis and improves the treatment outcomes and patient conditions.

## Figures and Tables

**Figure 1 antioxidants-11-01645-f001:**
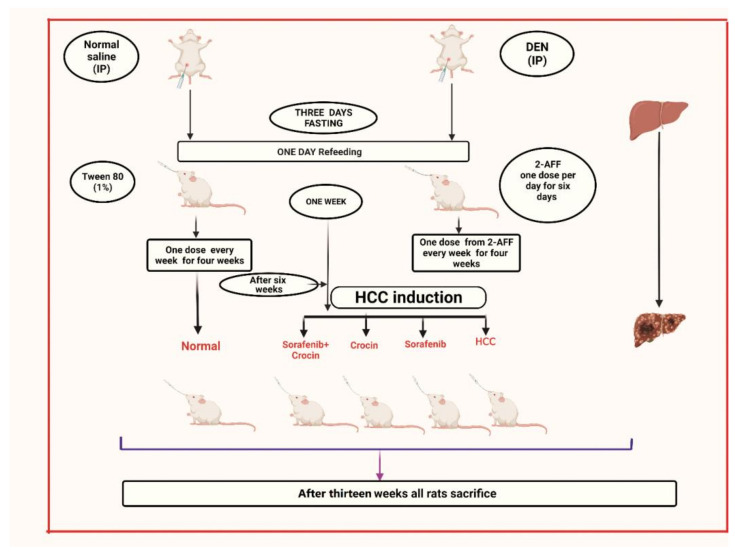
Schematic representation of the experimental setup followed for HCC induction and treatment.

**Figure 2 antioxidants-11-01645-f002:**
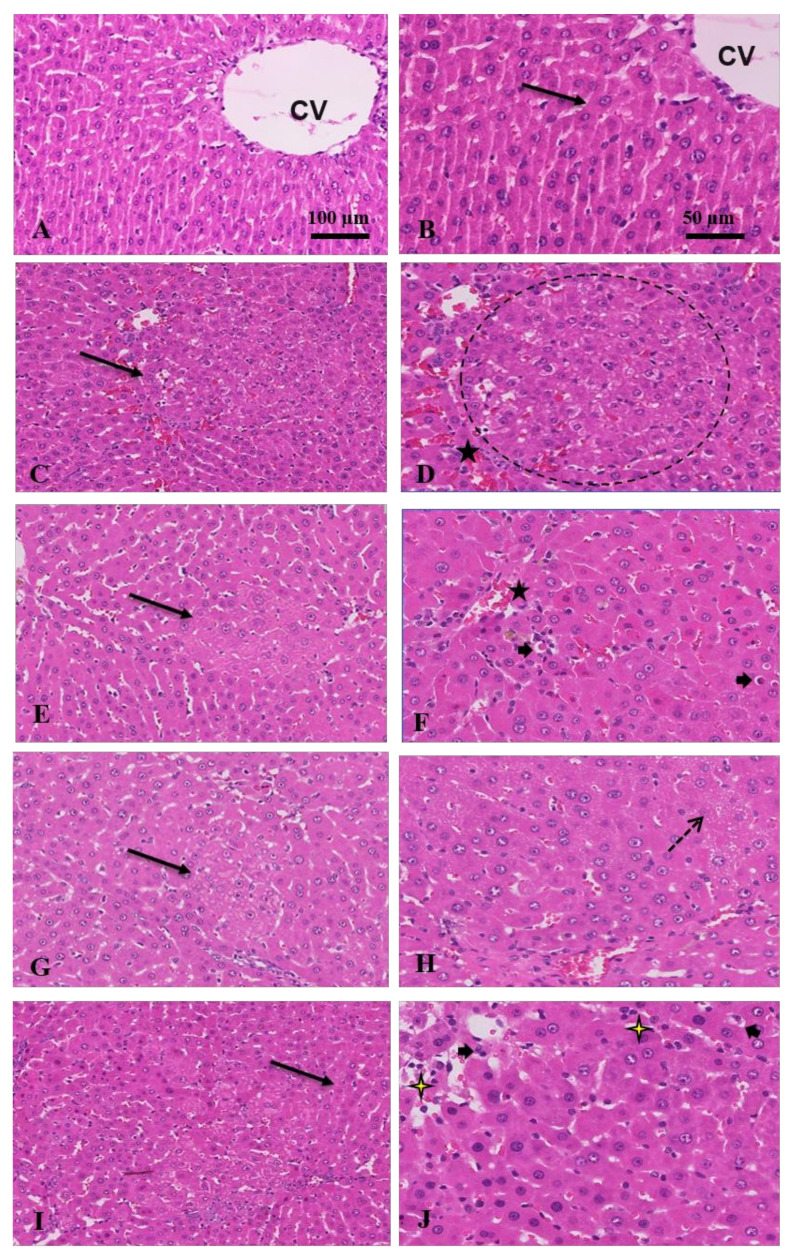
Anti-tumor effect of combination treatment: (**A**) control, untreated shows normal hepatocytes; (**B**) intact cell membrane (arrow), and central vein (CV); (**C**) HCC, a moderate increase in cell density (arrow); (**D**) wide distribution of hemorrhage (star) and tumor nodules (circle) with >3 cell strand thickness; (**E**) crocin, eosinophilic cytoplasm with marked reduction in tumor nodules (arrow), cellular density; (**F**) and hemorrhage (star), increase in apoptosis (arrowheads); (**G**) sorafenib, reduced tumor nodules (arrow); (**H**) with marked increased in fat vesicles or steatosis (dot arrow); (**I**) sorafenib and crocin, marked decrease in hyper cellularity, hemorrhage, and steatosis; normal strands thickness (arrow); (**J**) with increased apoptosis (arrowheads) and necrosis (yellow stars) (H&E, ×100 and 50 μm).

**Figure 3 antioxidants-11-01645-f003:**
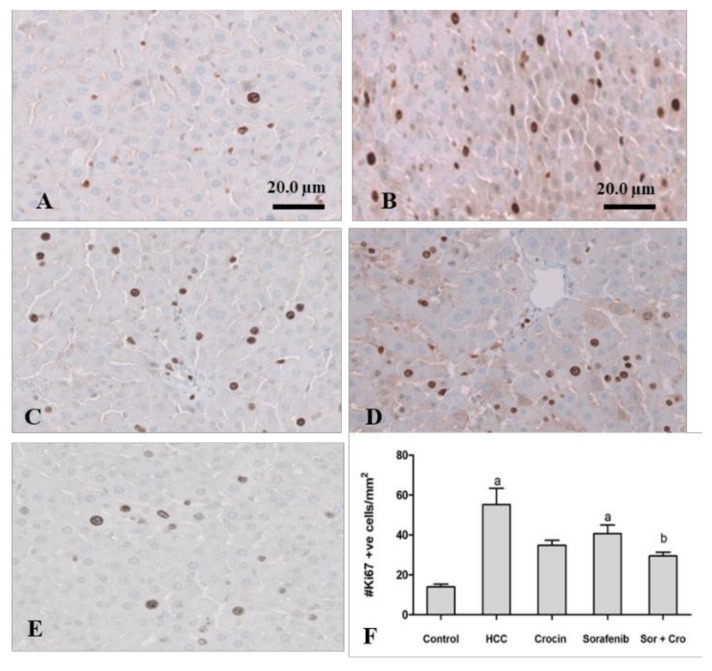
Antitumor effect of crocin and sorafenib on cell proliferation. Representative images of immunohistochemical staining with Ki-67 in control group (**A**). HCC group (**B**). crocin-treated group (**C**). sorafenib-treated group (**D**), and Crocin/sorafenib treated group (**E**). Panel (**F**) shows the quantitative analysis of Ki-67 immunoreactive cells in 10 fields of each section of the Ki-67 positive foci and quantitative region analysis of the Ki-67–positive foci ×100 magnification (Scale bar 20 µm). The values were evaluated by one-way ANOVA followed by Dunnett’s *t*-test compared to the HCC-induced group. Data are represented as mean ± SEM. The letter (a) is used for *p* < 0.05 vs. control group and the letter (b) is used for *p* < 0.05 vs. HCC group.

**Figure 4 antioxidants-11-01645-f004:**
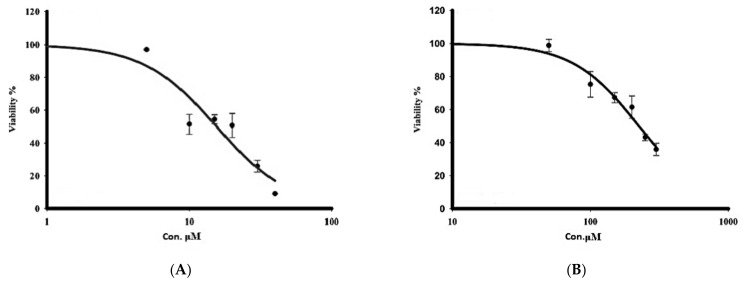
Effect of treatment with sorafenib (**A**) and crocin (**B**) on HepG2 cell viability. IC50 value corresponds to the concentration leading to a loss of 50% of cell viability. Data were expressed as mean ± SEM for three replicates (*n* = 3).

**Figure 5 antioxidants-11-01645-f005:**
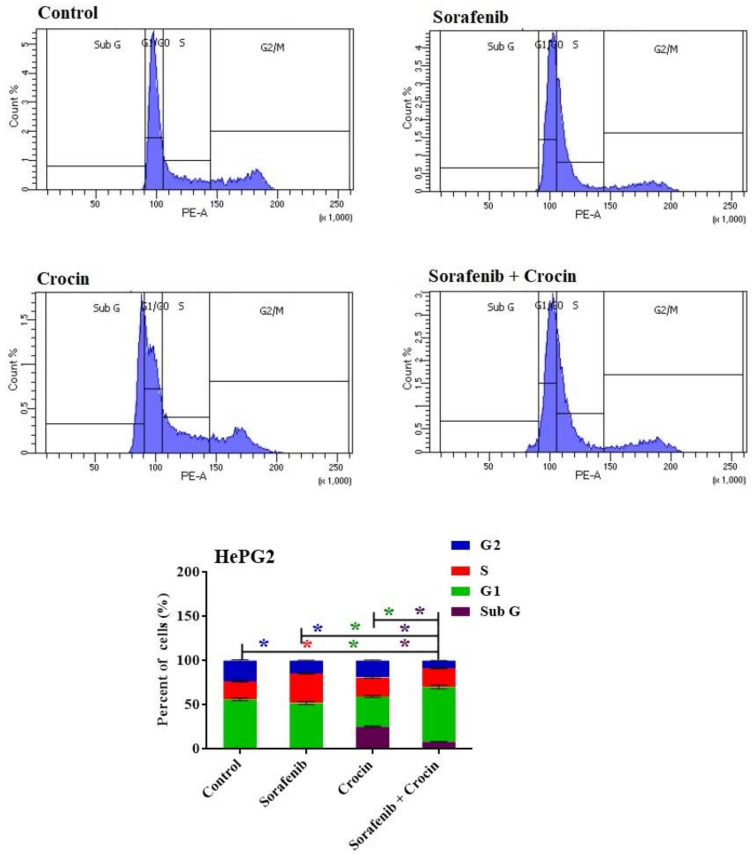
DNA cytometry analysis showing the effects of sorafenib and Crocin on cell cycle distribution of HePG2 cells. Cells were exposed to sorafenib, crocin and the mix for 48 h. Cell phases were plotted as percentage of total events. Sub-G cell population was plotted as percent of total events of cells. Data were presented as mean ± SEM for three replicates (*n* = 3). The differences of events from each respective control were considered significant at * *p* < 0.05.

**Figure 6 antioxidants-11-01645-f006:**
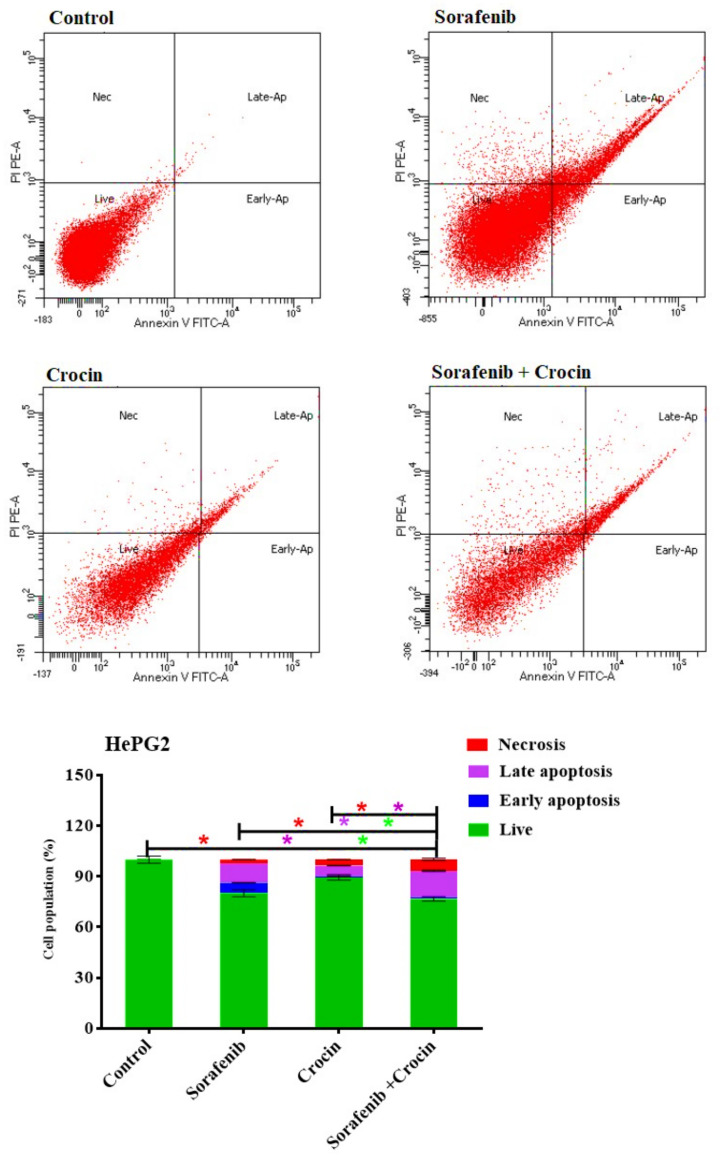
DNA cytometry analysis of Annexin V-FITC showing representative flow cytometry panels of apoptosis, necrosis, and cell vitality of HePG2 in response to 48 h exposure to media only (control) or (sorafenib, crocin, and (sorafenib/crocin), respectively). Cells population (%), including early apoptosis, late apoptosis, necrosis, and total cell death for the different treatments were shown in the lower panel. All data were presented as mean ± SEM for three replicates (*n* = 3). Differences were considered significant at * *p* < 0.05.

**Table 1 antioxidants-11-01645-t001:** Body weight (grams) and ratios of body weight changes (%) during 6 weeks after HCC induction in different studied groups.

Variables	Group 1 (Control)	Group 2 (HCC)
Total body weight at 1st week (grams)	189.30 ± 6.85	186.79 ± 11.69
Total body weight at 2nd week (grams)	189 ± 8.18	162.02 ± 9.17 ^a^
Total body weight at 3rd week (grams)	214.18 ± 12.78	175.18 ± 13.85 ^a^
Total body weight at 4th week (grams)	239.26 ± 22.16	203.02 ± 12.82 ^a^
Total body weight at 5th week (grams)	247.98 ± 25.03	224.33 ± 18.94 ^a^
Total body weight at 6th week (grams)	265.98 ± 31.38	239.39 ± 22.05 ^a^
Ratio of total body weight changes (%)	40.51%	28.16%

Data were expressed as mean +/− standard deviation. Significance was made using one-way ANOVA test followed by least significant test. a: significance versus Group 1. Significance at *p* < 0.050.

**Table 2 antioxidants-11-01645-t002:** Effects of the various treatments on the body and liver weights. Initial and final body weights, changes in body weight, liver weights and liver weight indexes are tabulated in all examined groups.

Data	Group 1 (Control)	Group 2 (HCC)	Group 3 (Induced HCC + Crocin)	Group 4 (Induced HCC + Sorafenib)	Group 5 (Induced HCC + Sorafenib/Crocin)
Initial body weights at week 7 (grams)	278.16 ± 29.56	247.19 ± 17.40 ^a^	248.08 ± 14.86 ^a^	263.74 ± 19.60	237.97 ± 22.82 ^a^
Final body weights at week 13 (grams)	319.58 ± 4.04	269.90 ± 33.36 ^a^	291.52 ± 18.96	316.55 ± 26.53 ^b^	287.43 ± 27.61
Ratio of total increase of body weight (%)	14.89%	9.19%	17.51%	20.02%	20.78%
Liver weights (grams)	8.78 ±1.31	7.98 ± 1.37	8.07 ± 0.59	8.50 ± 1.11	8.08 ± 0.88
Liver index (%)	2.75 ± 0.13	2.95 ± 0.29	2.78 ± 0.24	2.68 ± 0.22 ^b^	2.81 ± 0.13

Data were expressed as mean +/− standard deviation. Significance was made using one-way ANOVA test followed by least significant test. a: significance versus Group 1. b: significance versus Group 2. Significance at *p* < 0.050.

**Table 3 antioxidants-11-01645-t003:** Measured parameters in different studied groups.

Parameters		Group 1 (Control)	Group 2 (HCC)	Group 3 (Induced HCC + Crocin)	Group 4 (Induced HCC + Sorafenib)	Group 5 (Induced HCC + Sorafenib/Crocin)
Serum	ALT (U/L)	16.99 ± 3.98	79.80 ± 10.35 ^a^	21.00 ± 5.66 ^b^	37.00 ± 4.80 ^a, b^	32.00 ± 8.06 ^a, b^
	AST (U/L)	20.36 ± 4.72	107.10 ± 8.96 ^a^	26.00 ± 7.38 ^b^	55.20 ± 25.37 ^a, b^	45.76 ± 4.73 ^a, b^
	ALP (U/L)	43.80 ± 3.56	134.40 ± 14.47 ^a^	45.20 ± 6.34 ^b^	98.40 ± 22.07 ^a, b^	56.00 ± 7.28 ^b^
	TP (mg/mL)	6.91 ± 0.78	10.98 ± 0.95 ^a^	7.52 ± 0.76 ^b^	9.36 ± 0.48 ^a, b^	9.48 ± 0.49 ^a, b^
	Conjugated bilirubin (mg/dL)	0.31 ± 0.05	1.81 ± 0.31 ^a^	0.38 ± 0.13 ^b^	0.98 ± 0.08 ^a, b^	0.88 ± 0.15 ^a, b, c^
	TC (mg/dL)	121.20 ± 12.99	254.40 ± 35.56 ^a^	124.60 ± 8.08 ^b^	197.80 ± 11.58 ^a, b^	126.40 ± 6.88 ^b^
	TG (mg/dL)	73.80 ± 3.11	144.00 ± 38.97 ^a^	76.20 ± 4.15 ^b^	118.60 ± 9.37 ^a, b^	76.60 ± 4.67 ^b^
	CRP (mg/dL)	7.24 ± 1.27	26.60 ± 6.11 ^a^	10.92 ± 3.46 ^b^	19.38 ± 2.88 ^a, b^	7.56 ± 0.96 ^b^
	IL-6 (pg/mL)	4.99 ± 0.79	23.34 ± 5.53 ^a^	6.16 ± 1.43 ^b^	14.75 ± 1.77 ^a, b^	5.52 ± 1.03 ^b^
	LDH (U/L)	158.00 ± 19.20	448.40 ± 83.25 ^a^	214.00 ± 43.78 ^b^	337.20 ± 93.08 ^a, b^	207.60 ± 16.56 ^b, c^
	Serum tumor marker PIVKA-II (mAU/mL)	3.50 ± 0.55	17.34 ± 2.05 ^a^	7.34 ± 1.63 ^a, b^	13.36 ± 1.24 ^a, b^	8.53 ± 1.41 ^a, b^
Tissue homogenates	GSH (ng/mg proteins)	17.90 ± 5.37	2.08 ± 0.65 ^a^	10.84 ± 2.07 ^a, b^	5.82 ± 2.53 ^a, b^	12.04 ± 2.48 ^a, b^
	MDA (nmol/mg proteins)	0.40 ± 0.11	1.63 ± 0.31 ^a^	0.77 ± 0.23 ^a, b^	1.19 ± 0.23 ^a, b^	0.77 ± 0.17 ^a, b^
	Homogenate tumor marker AFP (ng/mg proteins)	13.64 ± 2.71	72.12 ± 10.87 ^a^	22.84 ± 2.18 ^a, b^	52.60 ± 10.16 ^a, b^	25.16 ± 4.61 ^a, b, c^

Data were expressed as mean +/− standard deviation. Significance was made using one-way ANOVA test followed by least significant test. ALT: Alanine aminotransferase; AST: Aspartate aminotransferase; ALP: alkaline phosphatase; TP: total protein; TC: total cholesterol; TG: triglyceride; CRP: C-reactive protein; IL-6: interleukin-6; LDH: lactate dehydrogenase; PIVKA-II: Protein induced by vitamin K absence-II; GSH: glutathione; MDA: malonaldehyde; AFP: *α* fetoprotein. a: significance versus Group 1. b: significance versus Group 2; c: significance versus Group 4. Significance at *p* < 0.050.

**Table 4 antioxidants-11-01645-t004:** Expression fold changes of TNF-a, VEGF, P53 and NF-KB in HCC induced rats and in HCC rats treated with crocin, sorafenib or a combination of both drugs.

Rats Group	Expression Fold Change 2^−(△△CT)^							
	TNF-α	*p*-Value	VEGF	*p*-Value	P53	*p*-Value	NF-KB	*p*-Value
Control	1	0.00292043	1	0.15289108	1	0.25597226	1	0.05884745
HCC	1083.8 ± 1.69	0.00558695	545.86 ± 2.91	0.02391665	243.43 ± 2.85	0.00896182	352.38 ± 3.86	0.00558695
HCC+ Crocin	1.17 ± 0.422	0.03644899	5.02 ± 1.58	0.4917508	0.99 ± 0.86	0.07553378	4.21 ± 2.48	0.08432088
HCC+ Sorafenib	165.79 ± 1.47	0.01215541	54.57 ± 1.05	0.01092501	147.64 ± 1.17	0.00276726	12.87 ± 1.47	0.01215541
HCC+ Sorafenib/Crocin	95.8 ± 1.77	0.04617572	31.2 ± 1.65	0.08639409	24.1 ± 1.82	0.01113539	9.29 ± 1.28	0.18709868

*p*-values are reported as compared to the GAPDH gene for the same group.

**Table 5 antioxidants-11-01645-t005:** Effect of adding an equimolar mix of sorafenib and crocin at various concentrations on the HepG2 cell viability.

Concentration of Sorafenib and Crocin Mix (μM)	Cell Viability (%)
0	100
6.25	67.30
12.5	46.94
25	36.46
50	27.91
100	20.8

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
