# Peer review of "Therapeutic Effects of Crocin Alone or in Combination with Sorafenib against Hepatocellular Carcinoma: In Vivo & In Vitro Insights"

_antioxidants, 2022, doi:10.3390/antiox11091645_

Round 1
Reviewer 1 Report
Abdu and colleagues analyzed the effect of crocin treatment in hepatocellular carcinoma by using a rat HCC model and HepG2 cells. It is well known that crocin has an ant- tumor effect in several tumor entities and also a limited number of crocin treatment in HCC models are available. Here they analyzed the single effect of crocin and also in combination with sorafenib.
The basics of hepatocellular carcinoma are decribed in detail in the introduction. On the other side information about crocin is limited in this chapter. I suggest to give some more detail about the anti-cancer effect of crocin in the introduction, also in other cancer entities.
There are hugh differences for the described chemically induced-HCC model through DEN and 2-AAF in the literature. Why do the authors decide to use the described model. Why a three days fasting protocol is necessary? Is the DEN/2-AAF treatment not sufficient to receive a mitotic proliferative stimulation? Feed deprivation for three days is considered as a severe stress and itself lead to body weight loss and other side effects.
Table 3, page 8: The final body weights are depicted at week 13 and initial body weights at week 7. Figure 1 shows that the experiments last 12 weeks. à 7 weeks start point plus 12 weeks would result to take final body weights at 19 weeks. This has to be explained is more detail, maybe with as a timeline.
There are large differences in the initial body weight of the rats (Table 3). Did the authors used the same gender for their experiments? Male and female rats would show differences in the induction of hepatocellular carcinoma. The use of both gender would lead to differences itself and would distort the results.
Figure 2A and B shows the same photograph but with a different magnification bar! No tumor nodule is vissible in Figure 2E and G. à Figures must be replaced.
Figure 3B is compressed à replace
Figure 4: Align Figure 4A and B and use same font size for both figures.
The manuscript must be checked for spelling and grammatical errors.
Author Response
Reviewer 1
Abdu and colleagues analyzed the effect of crocin treatment in hepatocellular carcinoma by using a rat HCC model and HepG2 cells. It is well known that crocin has an ant- tumor effect in several tumor entities and also a limited number of crocin treatment in HCC models are available. Here they analyzed the single effect of crocin and also in combination with sorafenib.
Question 1. The basics of hepatocellular carcinoma are decribed in detail in the introduction. On the other side information about crocin is limited in this chapter. I suggest to give some more detail about the anti-cancer effect of crocin in the introduction, also in other cancer entities.
Answer. A new paragraph was added in the introduction section (line 129) to address the use of crocin in cancer treatment: “Recent findings uniformly showed that saffron and its derivatives can affect carcino-genesis in a variety of in vivo and in vitro models. Particularly, crocin displayed a sig-nificant anticancer activity in breast, lung, pancreatic and leukemic cells (Samarghandian et al., 2013). In addition to its chemo preventive effects against HCC [26], crocin displayed anti-proliferation and anti-apoptotic effects on cancer HepG2 cell lines [27] and contributed to autophagic HCC apoptosis [28]. The anti-cancer effect of crocin was explained by many mechanisms including the interaction with telomeric quadruplex sequences and down regulation of hTERT expression. Data indicated that telomerase activity of HepG2 cells decreases after treatment with crocin, which is prob-ably caused by down‑regulation of the expression of the catalytic subunit of the enzyme (Noureini et al., 2012). Crocin was also found to induce apoptosis through chromatin condensation and DNA fragmentation in human pancreatic cancer cell line (BxPC‑3) (Bakshi et al., 2010)”. (The same paragraph was indicated for reviewer 1, question 1).
Question 2. There are hugh differences for the described chemically induced-HCC model through DEN and 2-AAF in the literature. Why do the authors decide to use the described model. Why a three days fasting protocol is necessary? Is the DEN/2-AAF treatment not sufficient to receive a mitotic proliferative stimulation? Feed deprivation for three days is considered as a severe stress and itself lead to body weight loss and other side effects.
Answer. To clarify this point, a new paragraph is added in the Results section (line 347): ”In the two-stage HCC induction model protocol, initiation and promotion steps are important in developing HCC where promotor induces clonal expansion of initi-ated cells (Santos et al., 2017). To improve HCC development in animal model, expo-sure to a tumor promotor, such as 2-acetylaminofluorene (2-AAF), often help inducing the formation of altered hepatocytes foci (AHF) and hyperplastic nodules that would ultimately develop into HCC (Tolba et al 2015; Santos et al, 2017). Fasting-refeeding and employing 2-AAF after 2 weeks of using DEN are reported as mitotic proliferative stimuli (Hayden and Ghosh, 2014). In the present model, initiation is followed by a mitotic proliferative stimulus (Fasting and re-feeding) during treatment with promot-ing agent such as 2-Acetyl Aminofluorene (2-AAF) that induces selective proliferation of the initiated cell population over non-initiated cells in the target tissue (Espandiari et al., 2005; Feo et al., 2007). Although Feed deprivation for three days is considered as a severe stress and itself may lead to body weight loss, it reduces the death that results from other models, such as cutting a piece of the liver, and we have established this model in our previous experiments to induce liver cancer (Hamza, et al., 2021; Abdalla et al., 2022)”.
Question3. Table 3, page 8: The final body weights are depicted at week 13 and initial body weights at week 7. Figure 1 shows that the experiments last 12 weeks. à 7 weeks start point plus 12 weeks would result to take final body weights at 19 weeks. This has to be explained is more detail, maybe with as a timeline.
Answer. HCC induction in rats lasted for 6 weeks. Starting from week 7 up to week 13, the anti-HCC treatment was applied for another extra 7 weeks. The duration of the total experiment is 13 weeks and not 19. Information in figure 1 is updated (after 13 weeks, rats were sacrificed).
Question 4. There are large differences in the initial body weight of the rats (Table 3). Did the authors used the same gender for their experiments? Male and female rats would show differences in the induction of hepatocellular carcinoma. The use of both gender would lead to differences itself and would distort the results.
Answer. We used only male Wistar rats in these experiments. In Table 3, there is an important difference in the body weight between group 1 (Normal rats) and those of the other groups (all HCC induced rats), This is due to the HCC induction that yielded a weight loss as discussed in the manuscript. Among the treated groups (group 2 to 5), differences in body weights are not significant. Small differences were noticed and are likely due to discrepancies in individual responses of rats to HCC induction.
Question 5. Figure 2A and B shows the same photograph but with a different magnification bar! No tumor nodule is visible in Figure 2E and G. à Figures must be replaced.
Answer. Figure 2 was edited according to reviewer’s remark.
Question 6. Figure 3B is compressed à replace
Answer. Figure 3B was replaced by a better one.
Question 7. Figure 4: Align Figure 4A and B and use same font size for both figures.
Answer. This was corrected in the revised version of the manuscript.
Question 8. The manuscript must be checked for spelling and grammatical errors.
Answer. The manuscript was checked, and spelling and grammatical errors are now corrected.

Reviewer 2 Report
Dear authors,
For starters I want to congratulate you for the article.
There are some negative aspects, which I believe need to be clarified and corrected in order to publish this manuscript. The chosen study is interesting and complex, however, I have the following remarks and questions:
- Line 22-23: ‘Serum inflammatory markers (C- reactive protein (CRP); interleukin- 6 (IL-6); lactate dehydrogenase (LDH)), and oxidative stress markers were significantly induced in HCC group and were restored upon treatment with either or both of therapeutic molecules’ – the word ‘induced’ should be replaced with ‘increased’
- Line 30, 31, 32, 93: misspellings
- Line 96: ‘Crocin (catenoid) is the most abundant polyphenol of saffron; stigmas of the flower of Crocus sativus’- Crocin is a carotenoid, not a polyphenol
- Line 110: It is not the correct way of citation (according to the specific writing instructions of the journal).
- Line 126: Where did you obtain the crocin and sorafenib from? This information is not mentioned in your article.
- Lines 131 and 137-138: You should rephrase these sentences for a better understanding.
- Lines 142-148: Please specify which types of kits and which types of methods were used in the analysis of the plasma parameters.
- Lines 152-154: Please specify which types of kits and which types of methods were used in the analysis of the plasma parameters. Was the total protein concentration analysis performed in the protein extracts?
- Starting at line 188: The source from which crocin and sorafenib were obtained, as well as what kind of solutions were prepared, must be indicated in the experiments on cell cultures (cytotoxicity and apoptosis).
- Line 260: 3.1. This Effect of crocin and sorafenib treatment on HCC induced rats’- You should delete the word ‘this’ in here
- Figure 4: Figures A and B are not equal in size; I consider that for a better visualization you should edit this
- Figure 7 needs editing. Maybe put the table separately or just make it smaller and increase the size of the other parts of the figure for a better understanding.
- Line 504: ‘The fact that sorafenib increased necrosis and apoptosis but did not induce cell cycle arrest is indicative that it acts through differs mechanisms.’- Could you elaborate? What mechanisms are you considering here?
Overall, the manuscript is complex and the information is valuable, but you could improve the way you expose your findings in order to ease the reading of your article and highlight the most important ones.

Author Response
Reviewer 2
Question 1. Line 22-23: ‘Serum inflammatory markers (C- reactive protein (CRP);
interleukin- 6 (IL-6); lactate dehydrogenase (LDH)), and oxidative
stress markers were significantly induced in HCC group and were restored
upon treatment with either or both of therapeutic molecules’ – the word
‘induced’ should be replaced with ‘increased’
Line 30, 31, 32, 93: misspellings
Answer. These errors are corrected in the revised version of the manuscript.
Question 2. Line 96: ‘Crocin (catenoid) is the most abundant polyphenol of saffron;
stigmas of the flower of Crocus sativus’- Crocin is a carotenoid, not a
polyphenol
Answer. The corresponding sentence was changed as follows (line 126): “Crocin is a catenoid derived from saffron; stigmas of the flower of Crocus sativus”.
Question 3. Line 110: It is not the correct way of citation (according to the
specific writing instructions of the journal).
Answer. The citation was corrected.
Question 4. Line 126: Where did you obtain the crocin and sorafenib from? This
information is not mentioned in your article.
Answer. It is now indicated (line 167) that “Sorafenib and crocin at 98% purity were obtained from Sigma Chemical Co., Missouri, USA”.
Question 5. Lines 131 and 137-138: You should rephrase these sentences for a better
understanding.
Answer. For a better understanding, the paragraph (line 174) was rephrased as follows:
“After one week of DEN treatment followed by six weeks of 2-AAF administration, rats were subjected to anti-cancer treatment during six weeks. At week 13, blood samples were collected through retro orbital punctures, and animals were then sacrificed.”
The sentence (line 204) was rephrased as follows: “The body weights of rats were weekly recorded.”
Question 6. Lines 142-148: Please specify which types of kits and which types of
methods were used in the analysis of the plasma parameters.
Answer. kits and methods used in the analysis of the plasma parameters were specified: (line 214): “… were all estimated using the rat ELISA kits obtained by BioSource USA following the instructions and steps contained in the internal kits bulletin.”
And line 223: “….using ELISA rat kits purchased from BioSource (USA) and following the kit instructions. Protein concentration of the above supernatant was estimated by method of Lowry et al. [37].”
Question 7. Starting at line 188: The source from which crocin and sorafenib were
obtained, as well as what kind of solutions were prepared, must be
indicated in the experiments on cell cultures (cytotoxicity and apoptosis).
Answer. This information is now indicated in Material and Methods section (line 167): “Sorafenib and crocin at 98% purity were obtained from Sigma Chemical Co., Missouri, USA.”
Question 8. Line 260: 3.1. This Effect of crocin and sorafenib treatment on HCC
induced rats’- You should delete the word ‘this’ in here.
Answer. This is now corrected.
Question 9. Figure 4: Figures A and B are not equal in size; I consider that for a
better visualization you should edit this.
Answer. This is now corrected.
Question 10. Figure 7 needs editing. Maybe put the table separately or just make it
smaller and increase the size of the other parts of the figure for a
better understanding.
Answer. This figure was edited according to the reviewer comment. The table was removed, and percentages indicated in this table were reported in the text.
Question 11. Line 504: ‘The fact that sorafenib increased necrosis and apoptosis but
did not induce cell cycle arrest is indicative that it acts through
differs mechanisms.’- Could you elaborate? What mechanisms are you
considering here?
Answer. This statement was clarified as follows (line 772): “The fact that sorafenib increased necrosis and apoptosis but did not induce cell cycle arrest is indicative that it acts through differs mechanisms such as inactivation of the RAF/MEK/ERK pathway HCC and receptor tyrosine kinases involved in tumor progression and angiogenesis (Wilhelm et al., 2004)”.
Question 12. Overall, the manuscript is complex and the information is valuable, but
you could improve the way you expose your findings in order to ease the
reading of your article and highlight the most important ones.
Answer. New paragraphs are now added to the manuscript to better explain the findings reported in this paper. This includes mainly interpretation of the anti-proliferative, pro-apoptotic and cycle arrest effects of sorafenib and crocin on HCC cells and possible mechanisms of synergism between the two drugs (please see newly added paragraphs).
-----------------------

Reviewer 3 Report
The authors report the therapeutic effect of crocin, alone or in combination with sorafenib, in the treatment of hepatocellular carcinoma through in vivo and in vitro experiments. Following these big concerns improve the manuscript.
1. There is a lack of practical introduction of crocin in cancer treatment in the Introduction.
2. It is not necessary to introduce the common sense knowledge of liver cancer in too much detail, and the current treatment progress should be emphasized.
3. DNE was used at 200 mg/kg in the chemically induced-HCC animal model, and the reference for this dose should be marked.
4. Table 1 is removed from the main text or placed in the supporting information.
5. Figure 3 needs to be adjusted to a suitable display with consistent proportions.
6. How an equimolar mix of sorafenib and crocin works should be explained in detail.
7. The literature on sorafenib arresting the cell cycle should also be reported, and it is recommended that the authors describe as not found that sorafenib arrests the cell cycle under your experimental conditions. In addition, the necessary discussion should be added against the published literature.
8. The layout of Figure 7 is too unreasonable.
9. From the apoptosis data, it should be discussed whether the effects of the two are additive or synergistic.
10. As an additional question, how did the authors rule out a serious color effect of Crocin in their experiments?
Author Response
Reviewer 3
The authors report the therapeutic effect of crocin, alone or in combination with sorafenib, in the treatment of hepatocellular carcinoma through in vivo and in vitro experiments. Following these big concerns improve the manuscript.
Question 1. There is a lack of practical introduction of crocin in cancer treatment in the Introduction.
Answer. A paragraph is now added in the introduction section (lines 129 and 134) to address the role of crocin in cancer treatment: “Recent findings uniformly showed that saffron and its derivatives can affect carcinogenesis in a variety of in vivo and in vitro models. Particularly, crocin displayed a significant anticancer activity in breast, lung, pancreatic and leukemic cells (Samarghandian et al., 2013). The anti-cancer effect of crocin was explained by many mechanisms including the interaction with telomeric quadruplex sequences and down regulation of hTERT expression. Data indicated that telomerase activity of HepG2 cells decreases after treatment with crocin, which is probably caused by down‑regulation of the expression of the catalytic subunit of the enzyme (Noureini et al., 2012). Crocin was also found to induce apoptosis through chromatin condensation and DNA fragmentation in human pancreatic cancer cell line (BxPC‑3) (Bakshi et al., 2010)”.
Question 2. It is not necessary to introduce the common sense knowledge of liver cancer in too much detail, and the current treatment progress should be emphasized.
Answer. Some details about cancer are removed from the introduction and a short paragraph dealing with current treatments was added (please see introduction section).
Question 3. DNE was used at 200 mg/kg in the chemically induced-HCC animal model, and the reference for this dose should be marked.
Answer. The reference was now updated.
Question 4. Table 1 is removed from the main text or placed in the supporting information.
Answer. Table 1 was removed from the manuscript.
Question 5. Figure 3 needs to be adjusted to a suitable display with consistent proportions.
Answer. Figure 3 was now adjusted.
Question 6. How an equimolar mix of sorafenib and crocin works should be explained in detail.
Answer. A sentence in the corresponding paragraph was added in Material and Methods to specify the concentration of each compound in the equimolar mix (Line 280): “The attached Cells were treated with different concentrations of crocin (100 μM, 150 μM, 200 μM, 250 μM and 300 μM) and Sorafenib (5 μM, 10 μM, 20 μM, 30 μM and 40 μM) or sorafenib/ crocin equimolar combination (final concentration of the mix is 6.25 μM, 12.5 μM, 25 μM, 50 μM or 100 μM) for 48 h”.
To better explain the effects of both compounds in the combination treatment, a short paragraph was added in the Results section (line 781): “Sorafenib, crocin and combined treatment significantly increased the number of necrotic and apoptotic cells. The cell death by apoptosis and necrosis confirmed the synergetic effects “.
Another paragraph was also added to the discussion (line 875): ”Combining sorafenib and crocin at an equimolar ratio was effective in reducing cell viability and increasing apoptosis as compared to the use of either dug alone, as it was re-ported for other sorafenib combined treatments (Morisaki et al., 2013). The combined effect is likely to be explained by a synergism between sorafenib known to inhibit tu-mor proliferation and increase apoptosis in HCC (Wilhelm et al., 2004) and crocin. While sorafenib targets mainly the RAF/MEK/ERK pathway and receptor tyrosine kinases involved in tumor progression and angiogenesis, a complementary synergetic action of crocin may occur through other mechanisms including the interaction with telomeric quadruplex sequences and down regulation of hTERT expression, resulting in a decreased telomerase activity of HepG2 cells (Noureini et al. 2012)”.
Question 7. The literature on sorafenib arresting the cell cycle should also be reported, and it is recommended that the authors describe as not found that sorafenib arrests the cell cycle under your experimental conditions. In addition, the necessary discussion should be added against the published literature.
Answer. A paragraph was added in the introduction section (line 85) to emphasize the role of sorafenib in arresting the cell cycle: ”It was reported that treatment with sorafenib had no effects on the cell cycle distribution of many tumor cell lines (HT29, Bax−/− HCT116, H460, and SKOV3) (Plastaras et al., 2007). Sorafenib however delayed the cell cycle G1 in PC3 and T98G malignant glioma cells (Plastaras et al., 2007; Jane et al., 2006). The ability of sorafenib to arrest the cell cycle may be related to Raf kinase inhibition. In irradiated cancer cell lines, sorafenib delayed the cell cycle G1 phase through modulating the expression of cyclin D1 and Rb genes (Plastaras et al., 2007). Genes involved in cell cycle regulation (CDC45L, CDC6 and CDCA5) were downregulated by sorafenib in human HepG2 and Huh7 HCC cell lines (Cervello et al., 2012). In thyroid carcinoma cells, sorafenib was reported to inhibit multiple intracellular signaling pathways such as RAF-MAP kinase, leading to an increased proportion of cells in subG1 peak, cell cycle arrest and the initiation of apoptosis (Broecker-Preusset et al., 2015). Furthermore, sorafenib induced cell cycle arrest and apoptosis in MV4-11, EOL-1 and NB4 Leukemia cell lines (Auclair et al., 2007; Zhang et al., 2018). Nevertheless, other studies reported that treatment of HepG2 HCC cell lines with sorafenib decreased the number of cells in G1 and increased cells in S phase (Liu et al., 2006). Seemingly, the effect of sorafenib on the cell cycle is depending on the cellular background.”
Question 8. The layout of Figure 7 is too unreasonable.
Answer. The layout of Figure 7 was adjusted.
Question 9. From the apoptosis data, it should be discussed whether the effects of the two are additive or synergistic.
Answer. A short paragraph was added in The Results section to explain the synergetic effect of crocin and sorafenib (line 781): “Sorafenib, crocin and combined treatment significantly increased the number of necrotic and apoptotic cells. The cell death by apoptosis and necrosis confirmed the synergetic effects “.
Another paragraph was also added to the discussion (line 875):” Combining sorafenib and crocin at an equimolar ratio was effective in reducing cell viability and increasing apoptosis as compared to the use of either dug alone, as it was re-ported for other sorafenib combined treatments (Morisaki et al., 2013). The combined effect is likely to be explained by a synergism between sorafenib known to inhibit tu-mor proliferation and increase apoptosis in HCC (Wilhelm et al., 2004) and crocin. While sorafenib targets mainly the RAF/MEK/ERK pathway and receptor tyrosine kinases involved in tumor progression and angiogenesis, a complementary synergetic action of crocin may occur through other mechanisms including the interaction with telomeric quadruplex sequences and down regulation of hTERT expression, resulting in a decreased telomerase activity of HepG2 cells (Noureini et al. 2012)”.
In addition to slight modifications in the paragraph (line 701): “The calculated combination index (CI) value 0.65 was lower than 0.8 indicating a synergism between the two drugs [36]. The combination of sorafenib/crocin at 1:1 molar ratio displayed a more potent inhibitory power of the cancer cell viability than the in-dividual drugs sorafenib and crocin. The calculated combination index (CI) value 0.65 was lower than 0.8 indicating a synergism between the two drugs [36]”.
Question 10. As an additional question, how did the authors rule out a serious color effect of Crocin in their experiments?
Answer. The low doses of crocin (50/100/150/200/250 and 300 μM) solubilized in 0.1% DMSO used in our experiments yielded a slight yellow color that did not disturb the experiments.

Round 2
Reviewer 1 Report
The authors adressed all my questions. I do not have further comments.
Reviewer 2 Report
I read the manuscript and noticed that the authors made the requested changes. The quality of the article resubmitted has increased.
Reviewer 3 Report
The author has addressed all my concerns.